# Endoscopic Skull Base Repair Strategy for CSF Leaks Associated with Pneumocephalus

**DOI:** 10.3390/jcm10010046

**Published:** 2020-12-25

**Authors:** Anda Gâta, Corneliu Toader, Veronica Elena Trombitaș, Annamaria Ilyes, Silviu Albu

**Affiliations:** 1Department of Otorhinolaryngology, University of Medicine and Pharmacy ‘Iuliu Hațieganu’, 400349 Cluj-Napoca, Romania; veronicatrombitas@gmail.com (V.E.T.); silviualbu63@gmail.com (S.A.); 2Clinic of Neurosurgery, National Institute of Neurology and Neurovascular Diseases, 41914 Bucharest, Romania; corneliutoader@gmail.com; 3Department of Radiology, Clinical Recovery Hospital, 400066 Cluj-Napoca, Romania; annamaria_ilyes_90@yahoo.com

**Keywords:** pneumocephalus, skull base, endoscopic repair, CSF leak, cranialization

## Abstract

Background: Cerebrospinal (CSF) fluid leaks with associated pneumocephalus (PNC) represent a condition bearing serious risks for the patient, with little data available in the literature. Reported success rates of endoscopic skull base repair are lower when PNC is associated than in the case of simple CSF leaks. The present study represents an analysis of our experience with endoscopic management of this condition. Methods: Records of patients with pneumocephalus and associated CSF leaks, who underwent endoscopic skull base repair, were reviewed. Demographics, history, etiology of PNC, size of defect, surgical approach, reconstruction technique and complications were evaluated. Results: Twenty patients with CSF leaks and PNC underwent endonasal repair by the senior author between 2005 and 2019. Defect size was larger than 15 mm in all cases. All patients presented either worsening of PNC under conservative treatment or tension PNC. First-attempt closure of the defect was successful in all patients (100%), with resolution of the pneumocephalus. One patient developed a synechia in the proximity of the frontal ostium, as a postoperative complication. The mean follow-up was 39 months (range: 15–94 months). Conclusion: The present study represents a proposed argument for earlier endoscopic endonasal treatment in patients presenting CSF leaks and pneumocephalus.

## 1. Introduction

Pneumocephalus (PNC) is caused by entrapment of air inside the cranial cavity, usually indicating a breach in the craniodural barrier. Most common etiologies include trauma, iatrogenic (associated with neurosurgical procedures, open or endoscopic skull-base surgery), intracranial gas-producing microorganisms, congenital defects, neoplasms, CSF shunt-related complications or even spontaneous defects [1,2,3]_._ Simple pneumocephalus is a more frequent entity, usually with a self-limited evolution contained by conservative measures [2]. However, nonresolving, worsening and PNC present without history of recent trauma or neurosurgical procedures is an indication of a persistent skull base (SB) defect through which air enters the cranial cavity. Moreover, when the volume of entrapped air is considerable, tension pneumocephalus develops and intracranial structures are compressed, leading to serious neurological complications or even death [3]. Such cases call for urgent repair of the SB defect.

As a result of the multifactorial etiology and diversified therapeutic approaches, there is no consensus regarding treatment of pneumocephalus other than recommendations based on observations from centers with a high incidence of this pathology. Endoscopic skull base repair has become the treatment of choice for CSF leaks, with success rates over 90%; however, when PNC is associated, success rates appear to be lower. The present case series exemplifies our experience regarding endoscopic endonasal skull base repair (ESBR) in patients with persistent pneumocephalus and CSF leak, following failure of conservative measures.

## 2. Materials and Methods

### 2.1. Study Design and Patient Selection

Approval of this study was granted by the ethics committee research council (number 197/25.05.2020). The current retrospective review includes medical records of patients with pneumocephalus who had undergone endoscopic endonasal repair, by the senior author, between 2005 and 2020. Records of all patients with CSF leaks undergoing endonasal repair were analyzed, and only patients with associated PNC were included in the study. All included patients were hospitalized in the neurosurgical department and were treated according to the hospital’s CSF leak treatment protocol. After obtaining the diagnosis of pneumocephalus on high resolution CT scans, patients with simple pneumocephalus were treated conservatively by 100% O^2^ therapy for 24 to 48 h, bed rest in Trendelenburg position, prophylactic antimicrobial therapy and lumbar drains and were advised to refrain from Valsalva or similar actions. Three-dimensional volume rendering (VR) of CT scans was obtained with the AW Volumeshare 5 software (General electric, Boston, MA, USA) by using the preset “Air Structure” Figure 1a,b and Figure 2, Patients who presented resolution of PNC under conservative measures were excluded from the study because they did not undergo immediate endoscopic repair. Careful monitoring for neurological deterioration and serial CT scanning were performed. For patients with tension pneumocephalus and CSF leak, conservative measures were impracticable, and endoscopic endonasal repair of the defect was performed after emergency relief of the intracranial pressure. Furthermore, persistent (for more than 7–10 days) or worsening PNC with CSF leak was also an indication for endoscopic surgery. Demographics, history, etiology of PNC, size of defect, surgical approach, reconstruction technique and complications were retrieved from the database, including only patients with a minimum one-year follow-up.

### 2.2. Surgical Approach and Reconstruction Technique

Preoperative assessment of the defect was performed on a thin sliced CT scan and MRI. After identification of the defect, the mucosa and displaced bone fragments were removed, and the defect was measured with a curette. During surgery, in cases with defects difficult to identify, after complete spheno–ethmoidectomy, topical 10% fluorescein was applied on the skull base for guidance (a cotton pad soaked with fluorescein was placed intranasally on the skull base for a few seconds. After removal of the cotton pad, fluorescein color shifted from orange to green where CSF was found). Reconstruction was performed using only autologous materials. A multilayer technique was employed: an inlay mucosa or fascia lata, septal cartilage to reinforce the first layer and fascia lata graft as overlay. Surgicel and gelfoam were used to secure the repair, and a fat graft was applied as a buttress for defects located in the lateral sphenoid recess and frontal sinus posterior wall. Perioperative prophylactic antibiotherapy was administered intravenously for 48 h and continued orally until removal of the nasal packing. The main cause for PNC was a persistent dural breach following cranialization (6 patients; Figure 1, Figure 2 and Figure 3), succeeded by neurosurgical tumor excisions via open approaches (5 patients) and craniofacial trauma (3 cases). Pneumocephalus was a complication of endoscopic sinus surgery in four cases and of microscopic transsphenoidal surgery for pituitary adenoma in 2 patients. Most defects were located in the frontal sinus (9 patients), and 3 patients presented a combined defect (frontal + ethmoid 2 patients; ethmoid + sphenoid one patient). The remaining 8 cases presented ethmoid (4) and sphenoid (4) skull base breaches. Follow-up of each patient consisted of CT scan in the 24 to 72 h following surgery, followed by examination with nasal endoscopy at 1, 3 and 6 months and biannually afterwards. Cases with no recurrence of PNC or CSF leak were considered successful after a follow-up period of a minimum one year.

## 3. Results

Between 2005 and 2019, a total of twenty patients (9 female and 11 male) with a mean age of 44.5 (range: 21–67) years were diagnosed with pneumocephalus and had undergone transnasal endoscopic repair by a single surgeon (S.A.; Table 1). The defect size was larger than 15 mm in all cases (range: 15–95 mm). Aside from the clear rhinorrhea present in all patients, headache was the main presenting symptom (16 patients), and the remaining four patients manifested mental status changes. None of the patients presented meningitis or any sign of intracranial infection. All patients were admitted to the neurosurgical department, and lumbar drains were placed routinely as part of the of the service’s ongoing treatment protocol for patients with active CSF leaks. One patient presented tension pneumocephalus and underwent surgical treatment without previous conservative measures (Figure 4, Figure 5 and Figure 6). The mean follow-up period was 39.15 months (range: 15–94 months).

A successful first-attempt repair by means of transnasal endoscopic surgery was accomplished in all patients (100%), with closure of CSF leak and resolution of pneumocephalus. Regarding postoperative complications, one patient developed a synechia of the frontal sinus neo-ostium, requiring revision surgery. One patient presented tumor recurrence and was treated in the neurosurgical department.

## 4. Discussion

Most frequently, pneumocephalus is a complication of trauma or a recent neurosurgical procedure [3,4]. There are two proposed mechanisms explaining the occurrence of PNC. The first one is the “inverted soda-pop bottle” incident, described by Lunsford and colleagues [5], in which the negative pressure caused by CSF leakage causes air to enter intracranially. The second hypothesis, proposed by Dandy et al. [6], compares the dural breach to a unidirectional valve through which air only enters the cranial cavity. The typical presenting symptoms of PNC include headaches, altered mental status, seizures and clear rhinorrhea [2,7]. Regarding imaging methods of diagnosis, MRI can be used to describe the location and degree of pneumocephalus. CT scan with 3D reconstruction is the imaging modality of choice for diagnostics, due to the higher availability, lower cost and high likelihood of detecting as little as 0.5 mL of air [2,8,9,10].

Before the emergence of modern imaging techniques, the incidence of PNC was reported to be under 1% in patients with craniofacial trauma, the diagnosis being made on the basis of a plain X-ray [11,12]. However, with the wide-spread use of CT scans, 7–9% of patients present intracranial air as a complication of cranio–facial trauma [2,13]. A notable imaging study by Reasoner et al. revealed a 100% incidence of pneumocephalus after supratentorial craniotomy in the first 2 days after the procedure. Moreover, 75% of patients presented different grades of PNC one week after surgery, and in some cases it persisted up to three weeks [8]. Fortunately, endoscopic sinus surgery (ESS) is rarely (<1%) followed by intracranial complications, making PNC as a result of intranasal surgery quite rare [14]. On the other hand, with recent advances in endoscopic skull base surgery (ESBS), more extensive interventions are performed, leading to tumor excision rates similar to open approaches [15]. Extended excisions lead to noteworthy dural defects, which are difficult to seal, and thus the rate of CSF leaks is higher. Banu et al. prospectively reviewed 526 patients undergoing ESBS mostly for tumor excisions, and out of 258 patients with intraoperative CSF leaks, 102 (39.5%) presented PNC. Moreover, 66.7% of patients with postoperative CSF leaks presented pneumocephalus compared with 38% of patients without this postoperative complication. Therefore, pneumocephalus represents a risk factor for CSF leaks, and the two entities are frequently associated, but PNC is not always indicative of a leak. An intriguing remark made by the author in [9] regards the intracranial pattern of air distribution, which might help differentiate “benign” from “suspicious” pneumocephalus. Air in the sellar and parasellar regions, interhemispheric fissure, perimesencephalic cistern and on the cranial convexity is more suggestive for CSF leak occurrence, while intraventricular and frontal air is less concerning. Furthermore, Banu et al. evaluated the distribution patterns and volume of the intracranial air and quantified them through a scoring system. Interestingly, a correlation between a high PNC score and the development of a postoperative CSF leak was demonstrated [9]. This leads us to believe that relating etiology with pattern and volume of PNC could raise awareness upon the existence of an active CSF leak.

Regarding risk factors for development of PNC, abnormal body mass index (BMI) is linked to higher incidence of postoperative CSF leak and meningitis [16]. Allensworth et al. proposed the hypothesis that lower BMI determines sparse or even interrupted CSF flow through the breach, leading to greater risk of pneumocephalus or ascending meningitis [17]. This hypothesis was not confirmed by Banu et al., who found the type of skull base tumor pathology to be the main risk factor [9].

Placement of lumbar drains is controversial, some authors stating that PNC can be worsened by lumbar drains as a consequence of intracranial negative pressure [15,18,19,20]. Different authors recommend the use of lumbar drains for a maximum of three days and with caution, encouraging passive gravity-dependent drainage, a low flow rate of a maximum 5 mL/hour or epidural drains connected to JP bulb suction [1,15,21,22]. In our study, lumbar drains were routinely placed as part of CSF leak treatment protocol and we were not able to quantify the association with worsening PNC.

The occurrence of postoperative pneumocephalus indicates a breach between the cranial cavity and the external environment, the mechanism being mutual with CSF leak formation. The difference between simple CSF leak and that associated with pneumocephalus is that in the case of the latter, PNC transfers contaminated air inside the sterile intracranial area. Although a higher incidence of intracranial infections has been observed in patients with postoperative CSF leaks, little is mentioned in the literature about the implications of pneumocephalus [15,16]. Guo et al. retrospectively reviewed 251 patients with pituitary adenoma subjected to pure endonasal transsphenoidal surgeries and found postoperative pneumocephalus to be an important predictor of intracranial infections [15]. Avoidance of postoperative diaphragmatic defects was found to be of great significance in avoiding complications [15]. In addition, even if immediate postoperative pneumocephalus can be avoided by simple repair of diaphragmatic defects, some authors advocate the need for rigid reconstruction with septal cartilage or a titanium plate to prevent invagination of the sphenoid mucosa [23,24]. Pneumocephalus can develop in the sella floor; there is a report of tension pneumosella occurring five years after endoscopic endonasal transsphenoidal surgery [24]. Consequently, CSF leaks with persistent PNC could bear greater risks than simple CSF leaks [13,19]. Our study population did not experience intracranial infections, and even though the number of patients was small and all patients received prophylactic antibiotics, early endoscopic repair could also be responsible.

Craniodural defects are considered to have an inadequate healing due to the incapacity of the dura to regenerate, the defect being covered by a single layer of mucosa [25]. Enjamel and Foy found a 30.6% overall risk of meningitis in patients with posttraumatic CSF leak and a cumulative risk exceeding 85% in 10 years [26]. Similar results were reported by Bernal-Sprekelsen et al., where 33.9% of the patients treated conservatively in their study developed bacterial meningitis, with one patient developing a fistula and meningitis nine years after the traumatic event [25].

There is no consensus regarding pneumocephalus treatment, possibly due to the variate etiologies; however, certain conservative approaches are generally accepted, namely bed rest, oxygen therapy, lumbar drainage, airtight nasal packing and maintaining the patient in the Trendelenburg position. In case medical measures are insufficient, surgical temporizing options for decompression include needle aspiration, burr holes, ventriculostomy and craniotomy [2,4,21]. A more unconventional method proposed by certain authors is tracheostomy for the deviation of nasal airflow from the skull base defect [19,27].

Although success rates of ESBR in CSF leak closure are over 90% [17,28,29], a study by Wise el al. [1], evaluating patients with pneumocephalus undergoing endoscopic repair, reported a lower success rate of only 72%. The discrepancy in results between our study and the similar study by Wise et al. [1] can be explained by the reduced number of patients included in these studies and the decreased complexity of the cases included in our study. None of our patients underwent endoscopic skull base tumor excisions or postoperative radiotherapy, and all tumors were benign. Moreover, there is increased difficulty in identifying the SB defect site if a CSF leak is not associated, and only 65% of the patients included in the latter study associated an active CSF leak compared to all patients included in our study. Curiously, the author in [1] proposed PNC itself as a risk factor for failure due to the pressure differential at the site of the leak, an assertion not supported by our results. Due to the small number of patients and the success rates of 100%, we were not able to provide a more complex statistical analysis. Despite the lack of large studies evaluating endoscopic treatment of SB defects associating PNC, our study’s results are similar to other reports from the literature that present ESBR as a viable treatment for CSF leaks paired with PNC [30,31].

Regarding defect size, Del Gaudio observed the inefficiency of conservative measures in defects larger than 15 mm, an observation supported by the present study, taking into consideration that all defects were larger than 15 mm and all patients failed conservative measures [19]. A particularity of our patient population is the high number of patients with PNC complicating cranialization. The high success rates of endoscopic repair prove this approach to be a better alternative to revision craniotomy, whilst avoiding the morbidity associated with this open procedure.

## 5. Conclusions

Even though there is no consensus regarding treatment of pneumocephalus, first-line conservative measures are usually adopted. However, CSF leaks with associated PNC are a particular entity due to greater risks for the patient to develop complications. This study has shown high success rates of ESBR in CSF leak closure with resolution of pneumocephalus, contradicting another similar study and providing an argument for early endoscopic skull base reconstruction, even if the defect is not very large, not to mention the advantage of avoiding the long-term risk of ascending meningitis. We consider these arguments to recommend early endoscopic intervention in patients with pneumocephalus accompanying CSF leaks, or at least to present them as an option for these patients.

## Figures and Tables

**Figure 1 jcm-10-00046-f001:**
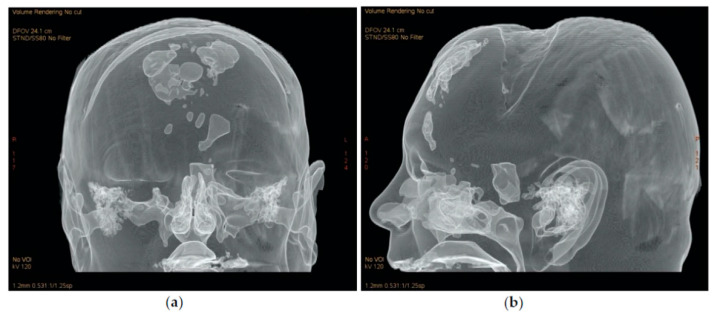
A 3D volume rendering of a CT scan of a patient (no. 12) with pneumocephalus (PNC) following cranialization. (**a**) Coronal view; (**b**) sagittal view.

**Figure 2 jcm-10-00046-f002:**
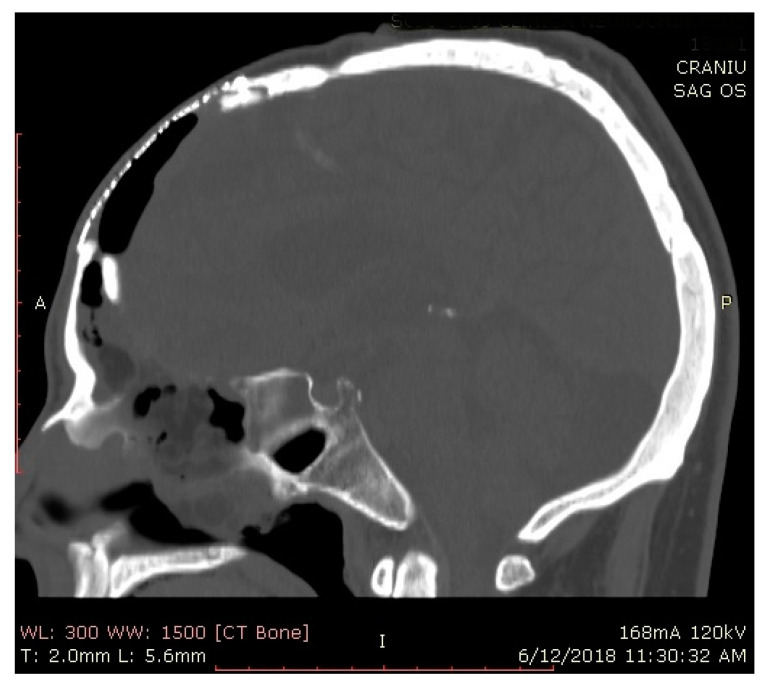
Cranial computed tomography, sagittal plane, of patient (no. 12) with PNC after cranialization.

**Figure 3 jcm-10-00046-f003:**
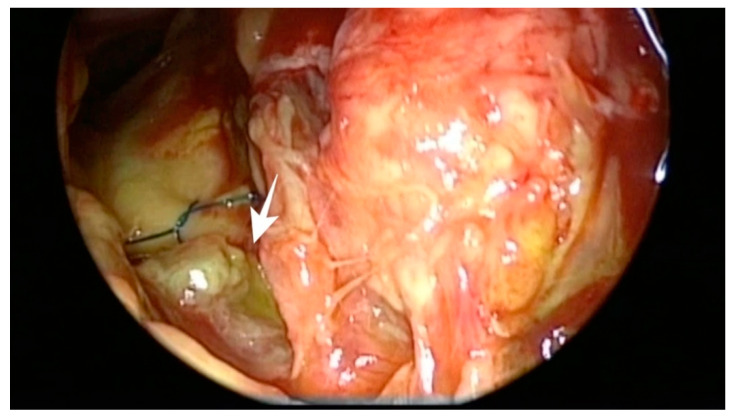
Endoscopic view of a CSF leak on the posterior wall of the frontal sinus, adjacent to the previous cranialization suture; white arrow points at the topic fluorescein shifting green in contact with CSF fluid (patient no. 12).

**Figure 4 jcm-10-00046-f004:**
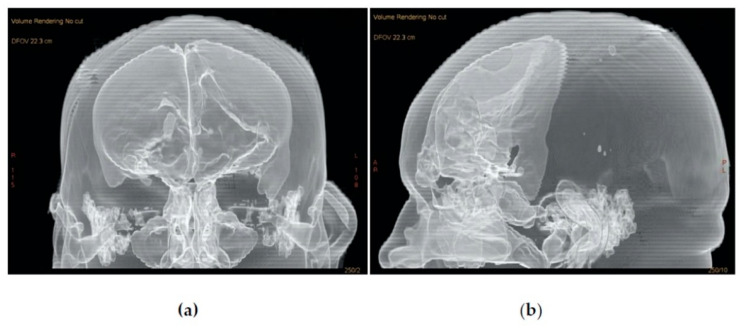
A 3D volume rendering of a CT scan of a patient (no. 6) with tension pneumocephalus. (**a**) Coronal view; (**b**) sagittal view.

**Figure 5 jcm-10-00046-f005:**
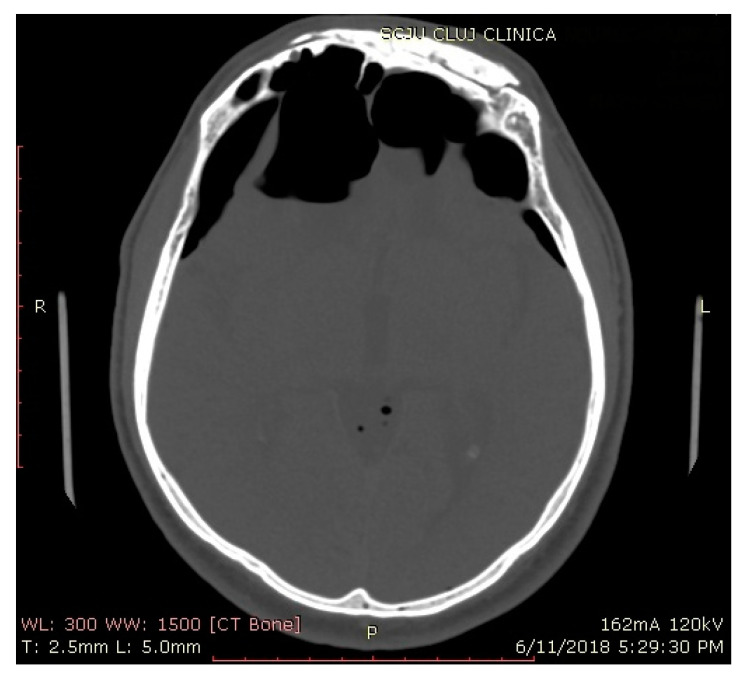
Cranial CT, axial plane, of a patient (no. 6) with tension pneumocephalus and visible Mt Fuji sign.

**Figure 6 jcm-10-00046-f006:**
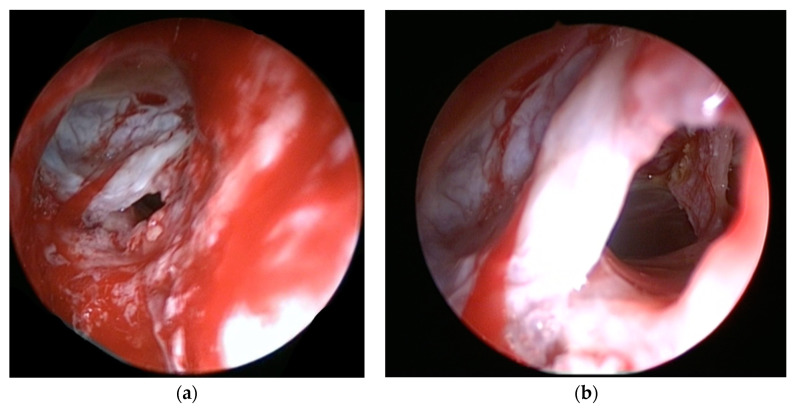
Endoscopic view of a significant bony defect and dural breach on the posterior wall of the frontal sinus (**a**), with intracranial endoscopic view (**b**) (patient no. 6).

**Table 1 jcm-10-00046-t001:** Characteristics of patients included in the study.

Patient	Age/Sex	Etiology	Defect Location	Defect Size (mm)	Complication	Follow-Up Months
1.	21/M	Trauma	Frontal p.w. + Ethmoid	75	no	17
2.	45/M	Sphenoid meningiom, transcranial approach	Sphenoid	50	no	94
3.	67/M	ESS	Ethmoid	25	no	72
4.	56/F	Cranialization	Frontal p.w.	15	no	55
5.	45/F	ESS	Ethmoid	35	no	36
6.	48/M	Cranialization	Frontal p.w. + Fr.	80	no	84
7.	65/F	Frontal meningioma, open approach	Fr. + frontal p.w.	60	surgery for tumor recurrence	32
8.	43/F	MTPS	Sellar diaphragm	30	no	24
9.	55/M	MTPS	Sellar diaphragm	20	no	18
10.	25/M	Cranialization	Fr.	25	no	25
11.	47/F	Sphenoid meningioma –transcranial approach	Sphenoid lateral recess	35	no	38
12.	35/M	Cranialization	Frontal p.w.	55	no	24
13.	29/M	Cranialization	Frontal p.w. + Fr.	70	neo-ostium stenosis	58
14.	39/F	ESS	Fr.	25	no	26
15.	61/M	Frontal tumor excision	Frontal p.w.	35	no	60
16.	55/F	Frontal tumor excision	Frontal p.w. + Fr.	85	no	17
17.	35/M	Cranialization	Fr. + ethmoid	60	no	48
18.	48/F	ESS	Ethmoid	30	no	18
19.	39/F	Trauma	Ethmoid	95	no	22
20.	32/M	Trauma	Sphenoid + ethmoid	40	no	15

ESS—endoscopic sinus surgery; MTPS—microscopic transsphenoidal surgery; Frontal p.w.—frontal sinus posterior wall; Fr.—frontal recess.

## Data Availability

The data presented in this study are available on request from the corresponding author. The data are not publicly available due to privacy reasons.

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
