# Peer review of "Endoscopic Skull Base Repair Strategy for CSF Leaks Associated with Pneumocephalus"

_jcm, 2020, doi:10.3390/jcm10010046_

Round 1

Reviewer 1 Report

The authors investigate the efficacy of the endoscopic skull base repair surgery for the treatment of pneumocephalus. This is an interesting paper. However, the study presents with some important issues and therefore a major revision should be required for the study to be eligible for publication:

  • The title/subject itself seems a bit inaccurate as the endoscopic repair aims on restoring skull base deficits that lead to clinical manifestations such as hydrocephalus, CSF leak, recurrent meningitis. Pneumocephalus itself is treated by means of conservative measures like bed rest and oxygen therapy(in the case of simple pneumocephalus) or surgery(in the case of pressure pneumocephalus.
  • There are some inconsistencies in the text. For example in the abstract section (line 19) and Results section (lines 86-87) the authors mention that all patients included in the study had defects larger than 15mm. In line 67 the authors state: “For small defects(<5mm) a free mucosal graft was used…”. Which is the case.
  • In line 54-55 the authors write: “Patients who presented resolution of PNC under conservative measures were excluded from the study.” Please clarify. Patients with pneumocephalus responding to conservative measures STILL have skull base deficits that should probably be addressed in the future.
  • In line 56-57 the authors mention that patients with tension pneumocephalus underwent direct endoscopic repair. How can endoscopic repair treat an emergency situation such as tension pneumocephalus that requires emergency craniotomy? Endoscopic repair can address the cause of pneumocephalus, NOT pneumocephalus itself.
  • In line 66 the authors write: “Fluoroscein was applied on the skull base for guidance”. This technique should be further clarified. The usual tactic included infusion of fluorescein through lumbar puncture before the operation and intraoperative identification of the breach. Please explain.
  • Lines 78-90 should be moved to the Materials and Methods section.
  • What is the rationale of placing lumbar drainage systems in all patients. A significant part of the literature disagrees with the placement of lumbar drains in patients with pneumocephalus and skull base deficits. Therefore, this decision should be further discussed.
  • Line 97: Tumor recurrence is not a complication of the endoscopic repair.
  • Line 130: “CT scan with 3d reconstruction”
  • Lines 144-145: This statement is not precise. The real problem here is the treatment of skull base deficits due to trauma or surgery and prevention of meningitis due to the latter. The literature addressing the time and strategy of surgical repair of such deficits is really extensive.
  • Linguistic errors appear throughout the text. For example:
    Line 123: Pneumocephalus is a complication of trauma or recent neurosurgery -> or recent neurosurgical procedure.
    Line 154: BMI determines sparse or even interruption of CSF flow -> BMI determines sparse or even interrupted CSF flow
    Line 185: …even though the number of patients is reduced -> …even though the number of patients is small.
  • The conceptualization of the study is somehow oversimplified. First, the fact that patients with large skull base defects need early surgical treatment is supported by many studies. The same applies for the sequelae of these deficits such as PNC and CSF leak. Second, the qualitative and quantitative analysis of the study is poor and should be expanded. Finally,a comparison of different endoscopic techniques and open techniques should be essential. 

Author Response

Response to Reviewer 1

Honored professor,

Thank You very much indeed for the time spent reviewing my paper. We have been immensely delighted with the possibility of publication in Your Journal. All comments and recommendations on Your part were appreciated. We are convicted that considering the points discussed and alterations made, this manuscript will have its quality increased. All changes requested were made on the manuscript using track changes and are highlighted in red in the text below - questions and responses.

  • The title/subject itself seems a bit inaccurate as the endoscopic repair aims on restoring skull base deficits that lead to clinical manifestations such as hydrocephalus, CSF leak, recurrent meningitis. Pneumocephalus itself is treated by means of conservative measures like bed rest and oxygen therapy(in the case of simple pneumocephalus) or surgery(in the case of pressure pneumocephalus.

 The title has been changed to ' Endoscopic skull base repair strategy for CSF leaks associated with pneumocephalus'.  Thank you for noticing this important inaccuracy, indeed the main scope of the article is focusing on CSF leaks associated with pneumocephalus. This conjunction represents a particular entity due to higher complication risk and lower success rate of endoscopic repair reported in literature. We now realize that the previous version of the title could have mislead the reader.

  • There are some inconsistencies in the text. For example in the abstract section (line 19) and Results section (lines 86-87) the authors mention that all patients included in the study had defects larger than 15mm. In line 67 the authors state: “For small defects(<5mm) a free mucosal graft was used…”. Which is the case.

    We apologize for this mistake, we utilize this type of reconstruction for CSF leaks with small skull base defects, however it does not apply to patients included in the present study. We appreciate the keen observation, proper corrections have been made (line 80).

  • In line 54-55 the authors write: “Patients who presented resolution of PNC under conservative measures were excluded from the study.” Please clarify. Patients with pneumocephalus responding to conservative measures STILL have skull base deficits that should probably be addressed in the future.

We agree patients responding to conservative measures still have indication for skull base repair. However, we chose to include only patients who associate both CSF leak and pneumocephalus, even if certain patients with resolution of PNC have latter undergone surgery. Further explanation was provided in the text- line 64.

  • In line 56-57 the authors mention that patients with tension pneumocephalus underwent direct endoscopic repair. How can endoscopic repair treat an emergency situation such as tension pneumocephalus that requires emergency craniotomy? Endoscopic repair can address the cause of pneumocephalus, NOT pneumocephalus itself.

Thank you for observing this confusing statement. We meant to underline that for patients with tension pneumocephalus, conservative measures were impracticable and endoscopic repair was provided along emergency actions such as needle aspiration and burr holes. Further explanation was provided in lines 66-67.

  • In line 66 the authors write: “Fluoroscein was applied on the skull base for guidance”. This technique should be further clarified. The usual tactic included infusion of fluorescein through lumbar puncture before the operation and intraoperative identification of the breach. Please explain.

Due to the risks associated with intrathecal use of fluorescein, some authors recommend the use of topic intranasal application of fluorescein for identification of the CSF leak site. This procedure implies that after the skull base is identified, a cotton pad soaked with fluorescein is placed intranasally, on the skull base for a few seconds. After removal of the cotton pad, fluorescein colour shifts from orange to green where CSF is found, and following the green stream upward, the breach can be identified. Even though more recent reports have demonstrated the safety of diluted low-dose intrathecal fluorescein, we have successfully been applying the topic intranasal method in our service for several years. The explanation on the use of topic Fluorescein was also added in the text ( lines 77-79)

  • Lines 78-90 should be moved to the Materials and Methods section.

Thank you for the recommendation, we moved the indicated lines from Results to Materials and Methods.

  • What is the rationale of placing lumbar drainage systems in all patients. A significant part of the literature disagrees with the placement of lumbar drains in patients with pneumocephalus and skull base deficits. Therefore, this decision should be further discussed.

It is the protocol of the neurosurgical department to place lumbar drains as a first-line treatment for all patients with active CSF leaks. Due to the fact that all patients were admitted in the neurosurgical department, evaluation of the ENT department was solicited after placement of lumbar drains. We corrected the reason for lumbar drain placement in the text. (line 58 and 201)

  • Line 97: Tumor recurrence is not a complication of the endoscopic repair.

Thank you for the observation, we have made the necessary corrections.

  • Line 130: “CT scan with 3d reconstruction”

Thank you for noticing, this feature was added to the text.

  • Lines 144-145: This statement is not precise. The real problem here is the treatment of skull base deficits due to trauma or surgery and prevention of meningitis due to the latter. The literature addressing the time and strategy of surgical repair of such deficits is really extensive.

We agree that the statement was not precise. Our intention was to exemplify that following endoscopic skull base surgery, pneumocephalus can be indicative of a CSF leak. We referrer to this etiology because our patient population included two subjects who underwent endoscopic removal of pituitary adenomas, classifiable as endoscopic skull base surgery. We rephrased this segment from discussions, making the information easier to follow and more explicit. (lines 164-182)

  • Linguistic errors appear throughout the text. For example: 
    Line 123: Pneumocephalus is a complication of trauma or recent neurosurgery -> or recent neurosurgical procedure. 
    Line 154: BMI determines sparse or even interruptionof CSF flow -> BMI determines sparse or even interrupted CSF flow
    Line 185: …even though the number of patients is reduced -> …even though the number of patients is 

       We appreciate the observations and have made the recommended corrections.

  The conceptualization of the study is somehow oversimplified. First, the fact that patients with large skull base defects need early surgical treatment is supported by many studies. The same applies for the sequelae of these deficits such as PNC and CSF leak. Second, the qualitative and quantitative analysis of the study is poor and should be expanded. Finally,a comparison of different endoscopic techniques and open techniques should be essential. 

   After intensive reviewing of the text we observed the possibility of a misunderstanding having occurred. Along the text we referred a lot to the etiology and treatment options for pneumocephalus. Our intention was to exemplify our results with treating only cases of PNC associated with CSF leaks. It is far from our scope to suggest treating all cases of pneumocephalus via endoscopic skull base repair. Furthermore we consider this entity to belong to the neurosurgical specialty. However, due to little reports found in literature on this subject and the high success rates found in our report we consider important to present this aspect. Due to the small number of patients and the success rates of 100% we were not able to provide a more complex statistical analysis. Nevertheless we consider important that our study contradicts data from literature which describes lower success rates for closure of CSF leaks if PNC is associated and provides an argument for early endoscopic skull base reconstruction even if the defect is not very large. We also added this explanation in the text- lines 247-248. We do not aim to compare endoscopic repair of CSF leaks to open approaches, the literature on the subject is extensive, and endoscopic repair has largely been proven as a more suited alternative, with better results and lower morbidity. We simply explained that an alternative to emergency cranialization for tension PNC, is simple needle aspiration or burr hole for emergent release of pressure, followed by endoscopic repair of the defect.

A new version of the manuscript, including the requested amendments, is being forwarded herein. Upon a careful reading of the manuscript, it became evident the improvements after reviewers’ analyses. If we have not reached the scope of the journal yet, we will be pleased to carry out any further modification(s). Thank You again and I remain sincerely yours.

Reviewer 2 Report

The manuscript “Endoscopic skull base repair strategy for pneumocephalus treatment” is a case series featuring 20 procedures of endoscopic endonasal skull base repair (ESBR) in patients with persistent pneumocephalus associated with CSF leak, following failure of conservative measures. The patients were operated in a timespan of 15 years (2005-2020) by the same surgeon (the Senior Author).

The background of the report is that pneumocephalus is a condition wherefore no standardized treatment protocols are available. Thus, the Authors discuss their usual conservative approach to the problem (which appears straightforward and is exhaustively explained) and the indications for skull base exploration and eventual repair (i.e. after failure of conservative therapy or in case of tension pneumocephalus). The results are clear and the discussion is coherent with them. The last section also provides a thorough review of the different aspects of diagnosis and treatment of pneumocephalus, maybe exceeding the scope of the article which should focus on the considerations arising from the case series itself. A summarization of the discussion is advisable.

The figures of the manuscript are good and the captions are plain. The table is complete, informative and well organised. Grammar and syntax along the text are good and provide enjoyable readability.

Based on the strengths and weaknesses of this work, I would propose minor revision.

Author Response

Response to Reviewer #2

Honored professor,

Thank You very much indeed for the time spent reviewing my paper. We have been immensely delighted with the possibility of publication in Your Journal. All comments and recommendations on Your part were appreciated. We are convicted that considering the points discussed and alterations made, this manuscript will have its quality increased. All changes requested were made and are highlighted in red in the text below - questions and responses.

  • The background of the report is that pneumocephalus is a condition wherefore no standardized treatment protocols are available. Thus, the Authors discuss their usual conservative approach to the problem (which appears straightforward and is exhaustively explained) and the indications for skull base exploration and eventual repair (i.e. after failure of conservative therapy or in case of tension pneumocephalus). The results are clear and the discussion is coherent with them. The last section also provides a thorough review of the different aspects of diagnosis and treatment of pneumocephalus, maybe exceeding the scope of the article which should focus on the considerations arising from the case series itself. A summarization of the discussion is advisable.
  • Thank you for underlining the necessity of summarization. We reordered the ideas presented in the Discussions section and we tried to make is more comprehensible for the reader. Also, we deleted certain information we did not consider significant for the scope of the paper.
  • Moreover, we thank you for noticing that the information provided in introduction could be improved. We added information on the subject trying to present what the study aims to underline.
  • Furthermore, the conclusion was improved, as recommended, and the idea that we propose early endoscopic treatment for PNC accompanying CSF leaks was reinforced.

A new version of the manuscript, including the requested amendments, is being forwarded herein. Upon a careful reading of the manuscript, it became evident the improvements after reviewers’ analyses. If we have not reached the scope of the journal yet, we will be pleased to carry out any further modification(s). Thank You again and I remain sincerely yours.

Response to Reviewer 2

Honored professor,

Thank You very much indeed for the time spent reviewing my paper. We have been immensely delighted with the possibility of publication in Your Journal. All comments and recommendations on Your part were appreciated. We are convicted that considering the points discussed and alterations made, this manuscript will have its quality increased. All changes requested were made and are highlighted in red in the text below - questions and responses.

The requested paper is discussed in the Discussion section, Reference 31, in red.

A new version of the manuscript, including the requested amendments, is being forwarded herein. Upon a careful reading of the manuscript, it became evident the improvements after reviewers’ analyses. If we have not reached the scope of the journal yet, we will be pleased to carry out any further modification(s). Thank You again and I remain sincerely yours.
